# Assessing the Quality of ChatGPT’s Dietary Advice for College Students from Dietitians’ Perspectives

**DOI:** 10.3390/nu16121939

**Published:** 2024-06-19

**Authors:** Li-Ling Liao, Li-Chun Chang, I-Ju Lai

**Affiliations:** 1Department of Public Health, College of Health Science, Kaohsiung Medical University, Kaohsiung City 807378, Taiwan; liling@kmu.edu.tw; 2Department of Medical Research, Kaohsiung Medical University Hospital, Kaohsiung City 807378, Taiwan; 3School of Nursing, Chang Gung University of Science and Technology, Tao-Yuan 333324, Taiwan; lichunc61@yahoo.com.tw; 4School of Nursing, College of Medicine, Chang Gung University, Tao-Yuan 333323, Taiwan; 5Department of Nursing, Linkou Chang Gung Memorial Hospital, Linkou 333423, Taiwan; 6Department of Nutrition, I-Shou University, Kaohsiung City 824005, Taiwan

**Keywords:** ChatGPT, dietary dvice, health education, nutrition literacy

## Abstract

Background: As ChatGPT becomes a primary information source for college students, its performance in providing dietary advice is under scrutiny. This study assessed ChatGPT’s performance in providing nutritional guidance to college students. Methods: ChatGPT’s performance on dietary advice was evaluated by 30 experienced dietitians and assessed using an objective nutrition literacy (NL) test. The dietitians were recruited to assess the quality of ChatGPT’s dietary advice, including its NL achievement and response quality. Results: The results indicate that ChatGPT’s performance varies across scenarios and is suboptimal for achieving NL with full achievement rates from 7.50% to 37.56%. While the responses excelled in readability, they lacked understandability, practicality, and completeness. In the NL test, ChatGPT showed an 84.38% accuracy rate, surpassing the NL level of Taiwanese college students. The top concern among the dietitians, cited 52 times in 242 feedback entries, was that the “response information lacks thoroughness or rigor, leading to misunderstandings or misuse”. Despite the potential of ChatGPT as a supplementary educational tool, significant gaps must be addressed, especially in detailed dietary inquiries. Conclusion: This study highlights the need for improved AI educational approaches and suggests the potential for developing ChatGPT teaching guides or usage instructions to train college students and support dietitians.

## 1. Introduction

In an era marked by the pervasive influence of digital technologies, the health and wellness landscape has undergone a notable transformation because of the emergence of artificial intelligence (AI)-driven platforms [1]. Among these advancements are conversational large language models (LLMs) such as ChatGPT, which have revolutionized public health education [2,3]. Notably, ChatGPT serves as a significant digital assistant, offering users a diverse range of information and personalized recommendations, including health advice customized to their specific needs and preferences [4].

Platforms such as ChatGPT have gained worldwide attention for their impressive performance in producing well-structured, logical, and informative responses [5]. The integration of AI into health counseling offers promising benefits for encouraging individuals to adopt healthier practices. In a medical study, experienced thoracic surgical clinicians assessed the feasibility of using ChatGPT for perioperative patient education regarding thoracic surgery. The results indicated that 92% of the responses met the qualification criteria, indicating ChatGPT’s potential feasibility for patient education [6]. Additionally, studies related to cancer education [4], dermatology education [7], diabetes education [8], or other diseases [9] found ChatGPT to be applicable for clinical health education in various evaluation indicators.

However, in another study assessing ChatGPT’s patient-education materials for implant-based breast reconstruction, it was found that although ChatGPT could generate materials more rapidly, its readability was poorer compared to official materials. Additionally, the ChatGPT-generated content had 50% accuracy with most errors being information errors [10]. Another study focusing on men’s health found ChatGPT-generated content to have worse understandability than traditional patient-education materials [11]. ChatGPT was also found to be better than Google Search for providing general medical knowledge but worse for medical recommendations [12].

In the realm of dietary education, ChatGPT-3.5 demonstrates potential as an efficient tool for renal dietary planning in patients with chronic kidney disease [13]. This underscores its potential despite its lower accuracy rate compared to ChatGPT-4.0. However, even ChatGPT-4.0 has produced errors, for example in creating menus for vegetarians [14]. One study delved into its credibility in providing dietary advice for individuals with food allergies and found that, while generally accurate, ChatGPT had a propensity to generate harmful dietary recommendations. Common errors often involve inaccuracies in food portions, calorie estimations for meals, or overall diet composition [15]. While AI integration, such as using ChatGPT, in health counseling shows promise for healthier practices, limitations such as quality and accuracy issues in specific contexts must be addressed for better outcomes.

During the transition to college, young adults frequently develop unhealthy eating habits, as shown by research in diverse regions, including Taiwan [16,17,18]. These habits increase the risk of rapid weight gain and nutritional deficiencies with potential long-term health consequences [19]. This makes it crucial to establish healthy eating habits [20]. Nutrition education becomes imperative in responding to these obstacles by employing diverse approaches to promote healthy eating habits and cultivating positive behaviors regarding nutrition [21].

With the rise of AI technology, college students are also beginning to utilize ChatGPT for learning. College students have expressed concerns about the quality and reliability of information sources; however, overall, they hold a positive attitude toward using ChatGPT [22]. Therefore, the use of this tool for nutrition education should be considered as a future intervention trend. Given the specialized knowledge and expertise possessed by dietitians in the nutrition field, their perspectives are invaluable for evaluating the efficacy and reliability of AI-generated dietary recommendations, especially in vulnerable populations such as college students.

This study’s primary objective was to evaluate ChatGPT’s performance in providing dietary advice to college students utilizing a comprehensive approach that included dietitians’ perspectives on nutrition literacy (NL) achievement, quality indicators, and an objective NL test. Multidimensional evaluation enables the assessment of dietary recommendations from both objective and subjective perspectives. It also identifies potential limitations and areas for improvement. These findings are expected to guide the development of AI-driven dietary counseling tools and emphasize the importance of expert perspectives in evaluating digital health interventions.

## 2. Methods

This study implemented a multidimensional evaluation methodology, as illustrated in Figure 1. The figure presents the objective and subjective assessments adopted in this study as well as the sources and acquisition methods of the data used for evaluating ChatGPT. The details are described in the following Section 2.1, Section 2.2 and Section 2.3.

### 2.1. ChatGPT Input and Data Sources

Given that NL significantly influences healthy dietary behavior, the NL indicators for college students in Taiwan were established by dietitians through the Delphi consensus process [23]. Subsequently, a scenario-based online program was developed and evaluated based on these corresponding indicators [24]. To explore the common dietary challenges faced by college students, this study employed five realistic scenarios from the program. The central hypothesis posited that students would turn to ChatGPT to assist in resolving these scenarios. To ensure that the prompts used in ChatGPT were suitable for nonprofessional individuals, this study recruited 20 students from nonmedical departments using convenience sampling. These participants had prior experience using any version of ChatGPT to seek answers. In the online survey, the participants were asked to envision themselves as protagonists within the scenarios and articulate how they would seek solutions from ChatGPT by recording their inquiry methods (prompts). The research team then analyzed the collected data, identifying the most prevalent prompts used by the participants when consulting ChatGPT (see Appendix A).

### 2.2. Response Generation with ChatGPT

In this study, ChatGPT-3.5 was chosen for use because of its status as a free version, making it more accessible as a tool for public health education and consultation compared to the paid versions. The “New Chat” function was used to input questions sequentially and independently to facilitate each question’s processing. Subsequently, the responses generated by ChatGPT were recorded and documented. The Appendix A include prompts and their corresponding ChatGPT-generated responses for the five scenarios.

### 2.3. Assessing ChatGPT’s Response and NL Test Performance

Thirty dietitians meeting the following specific eligibility criteria were invited to participate: (1) possessing a valid Taiwanese dietitian license and (2) having at least three years of experience in nutrition counseling. These participants were recruited through the personal networks of the research team members using a snowball sampling method. An online survey was conducted to evaluate the responses generated by ChatGPT. The questionnaire consisted of three parts. First, the participants assessed the achievement level of the NL indicators by rating the extent to which ChatGPT responses in various scenarios aligned with the corresponding NL indicators for college students. The ratings ranged from “not achieved at all” to “partially achieved” and “fully achieved”. Second, seven criteria used in previous studies evaluating online health information [25] were selected according to ChatGPT characteristics to assess the ChatGPT response quality. These criteria include accuracy (whether a source or information is consistent with agreed-upon scientific findings), currency (whether a source or information is up to date), completeness (whether necessary or expected aspects of a subject or topic are provided), understandability (whether a source or information has appropriate depth, quantity, specificity, and error-free), readability (whether the information is presented in a form that is easy to read), relevance (whether the information is relevant to the topic of interest or information seekers’ situation and background), and practicality (whether the information can be readily applied by an individual). The respondents rated each criterion on a Likert scale ranging from 1 to 10 with higher scores indicating better performance. The ratings greater than 7 were labeled as “acceptable”. Finally, the participants provided open-ended feedback expressing their appreciation and concerns regarding the ChatGPT responses. They also shared opinions on using ChatGPT for nutritional education among college students.

To further assess ChatGPT’s ability to provide nutrition education, this study employed a published test designed to evaluate the NL of Taiwanese college students [26]. The test questions were presented to ChatGPT, and the proportion of correct answers given by ChatGPT was compared with the results of previous studies on Taiwanese college students’ NL.

### 2.4. Statistical Analysis

Due to the small sample size of this study and the focus on understanding the evaluation performance, the collected data were subjected to descriptive statistical analysis. Both parametric and nonparametric methods were employed to provide a comprehensive view of the data. Parametric measures such as means and standard deviations were used to illustrate central tendencies and variability, while nonparametric measures such as medians and ranges were included to highlight the diversity of evaluations among the participants. Additionally, the distribution of ChatGPT’s performance ratings was illustrated using a stacked bar chart and a line chart to provide a clear visual representation of the data. The correct rate (%) of ChatGPT’s ability on the NL test is also demonstrated. The perspectives of the dietitians regarding ChatGPT were organized, and their evaluation of the information was summarized. Additionally, the NL test accuracy rates were compared between ChatGPT and college students.

## 3. Results

### 3.1. The Achievement Level of NL Indicators

As presented in Table 1, the participants evaluated the achievement of NL indicators in dietary advice provided by ChatGPT across five scenarios. Among the corresponding indicators in each scenario, the fully achieved rates ranged from 7.50% to 37.56%. Conversely, in situations where achievement was not attained at all, the percentages ranged from 20.00% to 63.33%. This suggests that the participants generally believe there is room for improvement in ChatGPT’s response content to meet the expected corresponding NL indicators. Additionally, in terms of indicator achievement, Scenario 4: Fewer processed foods showed the best performance (fully achieved: 35.56%; not achieved at all: 20.00%).

### 3.2. Evaluation of Response Quality

Table 2 and Figure 2 and Figure 3 show that, among the seven evaluation indicators, readability consistently performed well across various scenarios (mean: 7.97–8.27, median: 8). The acceptability ratings (>7) fell within the 67%–80% range. In contrast, the three indicators with relatively poor performance were understandability (mean: 6.23–7.40, median: 7–8), practicality (mean: 6.30–7.67, median: 7–7.5), and completeness (mean: 6.70–7.90, median: 6.5–8). These three indicators received acceptable ratings with understandability and practicality having ratings not exceeding 50% in the four scenarios and completeness having ratings below 50% in three scenarios.

Table 3 shows that the participants provided 242 feedback entries regarding the use of ChatGPT for nutrition education among college students. Among these, positive opinions occurred 87 times (35.95%), whereas concerns occurred 155 times (64.05%). Positive opinions that appeared more than 20 times include “A02 Provides popularized, preliminary, and easy-to-understand dietary advice” (29 times) and “A04 Provides comprehensive and detailed health education information” (21 times). For the former, the dietitians mostly believe that ChatGPT’s responses can “*provide basic dietary concepts and suggestions using simple replies*” (#4, representing dietitian No.4, and the following as well) or “*quickly convey needed nutritional knowledge for the general public*” (#9). Regarding the latter, the dietitians stated that the information is “*sufficiently clear*” (#19) and “*can rapidly organize usable educational information with high completeness*” (#17).

In the concern category, “B02 Provides health education information lacking thoroughness or rigor, leading to misunderstandings or misuse” appeared 52 times, making it the most frequent concern. Related suggestions include “S5: Suggesting homemade fruit juice as a thirst-quenching method, but it may lead to a higher risk of hyperlipidemia or fatty liver in the long run” (#21), “S1: When mentioning lean meat, there is no specific clarification on which part of beef or pork is considered lean meat, which may cause misunderstanding” (#14), and “S3: Due to the lack of clear guidance on nutritional labeling, sodium recommendations, or nutritional claim regulations, readers still do not understand which products to choose after reading” (#1). The second most frequent concern is “B07 Health education information is incorrect, especially regarding food categorization” (23 times). The dietitians pointed out various errors in ChatGPT’s responses, such as “S1: Food categorization does not include the category of ‘legumes’” (#31), “S4: Pumpkin is not classified as a vegetable” (#13), “S3: Grilled meat involves the use of barbecue sauce, and it should not be listed as a healthy option” (#14), and “S4: The option of ham in high-quality protein choices is contradictory to the content of reducing processed food intake, as it is a high-calorie and high-sodium food” (#14). Another concern mentioned 20 times is “B06 Unable to provide personalized (dietary or exercise) analysis and recommendations”. The dietitians believe that ChatGPT’s responses have limitations in providing personalized recommendations, as expressed in comments like “Lacks personalized dietary analysis and recommendations” (#5), “Does not provide recommendations based on individual blood biochemical values, which may cause harm to the body” (#6), and “Can only provide very basic nutritional directions and does not adequately address individual needs” (#31).

### 3.3. ChatGPT’s Performance on the NL Test

As indicated in Table 4, the NL test used here comprises 32 items. A survey conducted among college students in Taiwan [26] revealed an average correct score of 77.40%. In comparison, ChatGPT exhibited an accuracy rate of 84.38% (27 out of 32 items correct), surpassing the NL level of Taiwanese college students. That said, among the incorrectly answered questions, the percentage of correct answers among the college students in Taiwan ranged from 54.2% to 73.6%. This suggests that ChatGPT did not perform well, even regarding moderately difficult questions.

## 4. Discussion

This is the first study to evaluate the quality of ChatGPT-generated diet-related responses using a multidimensional evaluation approach. The primary findings revealed that, while ChatGPT is proficient in providing prompt dietary guidance, it exhibits variable performance when evaluated by registered dietitians. These findings underscore AI’s potential as a supplementary tool in nutrition education but also emphasize the need to enhance its performance to manage nuanced and personalized dietary inquiries.

Regarding the dietitians’ perspective on ChatGPT’s NL achievement, the rates for fully meeting the NL indicators were all below 40 percent with significant variability across different scenarios. This indicates ChatGPT’s inconsistency and relatively low effectiveness in providing fully adequate educational information depending on the dietary scenario being addressed. NL education is an innovative element in nutrition education today that enhances individuals’ ability to make healthy dietary decisions [27]. Effective NL intervention can significantly alter college students’ dietary behaviors [24]. Consequently, an increasing number of nutrition education programs have been using enhancements in NL as their main intervention strategy [27,28,29].

While using ChatGPT-3.5 as a source of health education information, this study found that it could only provide corresponding nutrition knowledge responses. It has yet to enhance the level of its educational approach. These results suggest that future AI design and training should actively integrate multiple health behavior theories to offer additional specific operational suggestions for healthy dietary behaviors [30]. Currently, because ChatGPT can only serve to provide educational information, the NL-related education strategies should be provided by dietitians or health-related educators. Therefore, it is important to develop courses and guidelines to deliver NL intervention strategies.

Furthermore, among the seven indicators established here, the dietitians found that ChatGPT’s responses performed best for readability, whereas its understandability, practicality, and completeness were the least effective. These outcomes were similarly reflected in the dieticians’ feedback, which noted that, despite the generalized nature of ChatGPT responses, they often lacked the specificity and depth required for individual dietary counseling. Despite its strengths, a higher proportion of concerns were raised regarding the use of ChatGPT in nutrition education. Numerous studies have indicated that the materials produced by ChatGPT are highly readable [6,7,31], making this a likely ChatGPT characteristic. It can present information in an easily readable and organized manner, making it more accessible and absorbable.

In professional health education, however, nutrition counseling is specialized. Dietitians work as licensed healthcare professionals tasked with safeguarding public dietary health. In this context, the most crucial aspect is to tailor dietary plans to individual health needs [32]. This study found that, as most inquirers did not adequately disclose their personal health conditions to ChatGPT, the responses produced did not suit individual physiological characteristics. This poses a significant risk in nutritional counseling. Over one-third of the dietitians’ concerns about ChatGPT involved its “*lacking thoroughness or rigor, leading to misunderstandings or misuse*”. Consequently, dietitians often believe that such information cannot be adequately understood or utilized without professional assistance, a finding consistent with previous research [33,34,35].

This study employed prompts that closely mirror those typically used by college students when querying ChatGPT. However, these prompts may not adhere to the established principles of crafting effective ChatGPT prompts [36], which significantly influences response quality. Given that ChatGPT has become a prevalent source of information for college students [22], a critical need exists to develop and implement educational programs that teach college students how to effectively use it as a source of health information. Simultaneously, the awareness of ChatGPT’s limitations must be enhanced [4]. This approach will not only enable ChatGPT to serve as a supplementary tool in health education but also mitigate the risks associated with using it. Additionally, future research could examine the correlation between the quality and depth of prompts and the quality of health education information generated by ChatGPT, which would further underscore the importance of such educational programs.

The results from the objective NL test also suggested that, although ChatGPT generally surpasses average college students regarding NL knowledge, accuracy issues exist. The dietitians in this study highlighted that ChatGPT’s responses, whether in food categorization, portion calculation, or even the use of professional terminology, have accuracy problems. These results indicate that, without professional oversight, ChatGPT has substantial limitations in practical application. Compared to previous studies, while the NL test accuracy rate was not inferior to that of other professional medical assessments [37,38,39], the test was primarily set at a basic difficulty level [26]. Better performance is expected, as in the perfect scores observed in other studies such as the diabetes knowledge questionnaire (24-DKQ) [8]. At this stage, ChatGPT’s role should be supportive. Considering the heavy workload of dietitians [40], establishing a practical model in which ChatGPT assists in completing dietitians’ tasks is a crucial objective. Furthermore, considering that young students might have a more open attitude towards AI, incorporating AI education into dietitian training curricula at universities should be a future trend. Therefore, designing and evaluating effective AI-assisted dietitian training programs is also a priority for future research.

### Limitations

This study has several limitations that warrant consideration. First, the use of predefined scenarios may not encompass the broad range of dietary challenges faced by college students. This may affect our findings’ generalizability. Second, the small sample size and potential lack of diversity among the participants limit the scope of the conclusions. Additionally, the potential misunderstanding of this technology by the dietitians may have introduced bias into their responses. The results revealed that the dietitians’ evaluations of the same quality indicators were not concentrated, suggesting variability in their familiarity and proficiency with this tool. Future research should further assess the dietitians’ use of ChatGPT to build on these findings and address the limitations identified in this study.

## 5. Conclusions

This study offers a pioneering evaluation of ChatGPT’s performance in providing dietary advice to college students. The findings reveal that, while ChatGPT demonstrates proficiency in providing quick and accessible information, it falls short in delivering personalized, in-depth dietary counseling, which is essential for addressing unique nutritional needs. The inconsistency in the achievement of NL indicators and variability in response quality across different scenarios underscore AI technology’s current limitations in adapting to complex dietary inquiries.

Moreover, feedback from the dietitians highlighted significant concerns regarding the accuracy of ChatGPT responses, especially in areas requiring precise knowledge, such as food categorization, portion sizes, and the use of professional terminology. These shortcomings emphasize the crucial role of professional oversight in integrating AI tools into nutritional education to ensure that the advice provided is accurate and safe.

Despite these challenges, AI’s potential to support dietary practices cannot be overlooked. ChatGPT has shown capabilities that, if further developed and refined, can significantly enhance the efficiency and reach of nutritional counseling, particularly in settings burdened by high client volumes and limited resources. Future research should focus on enhancing the personalization capabilities of AI systems, improving the understanding of complex nutritional concepts, and seamlessly integrating these tools into professional healthcare practices.

In conclusion, AI, such as ChatGPT, has the potential to become a valuable tool in nutrition education. Its current application, however, should be approached with caution, ensuring that it complements, rather than replaces, the nuanced judgment of skilled dietitians. AI’s evolution in dietetics promises a future in which technology and human expertise collaborate to more effectively foster healthier dietary behaviors.

## Figures and Tables

**Figure 1 nutrients-16-01939-f001:**
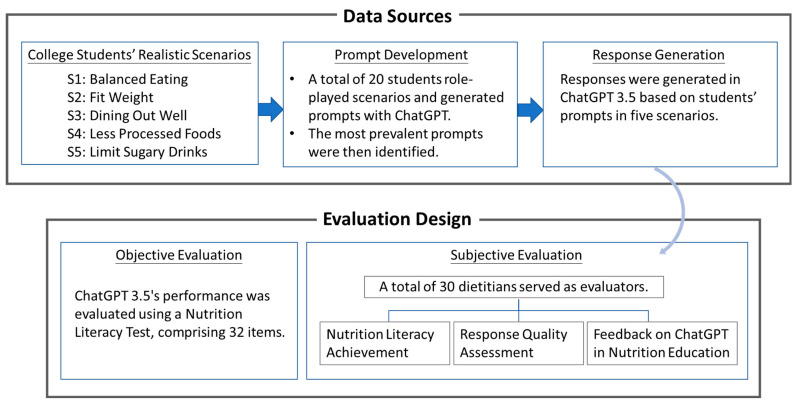
Overview of the multidimensional evaluation methodology.

**Figure 2 nutrients-16-01939-f002:**
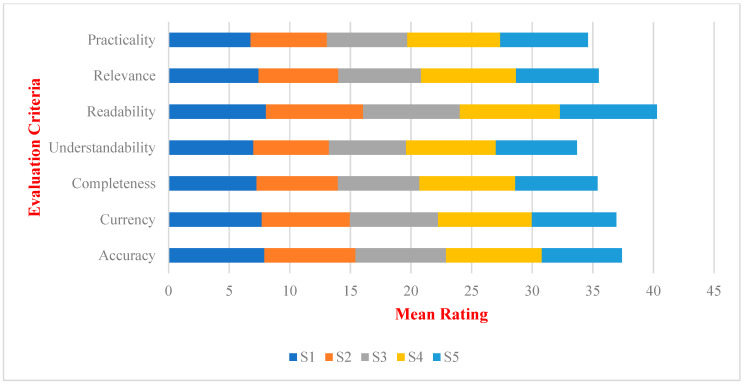
Comparison of ChatGPT’s mean performance ratings in dietary advice for each criterion. Note: scenarios: S1: Balanced Eating; S2: Fit Weight; S3: Dining Out Well; S4: Fewer Processed Foods; S5: Limit Sugary Drinks.

**Figure 3 nutrients-16-01939-f003:**
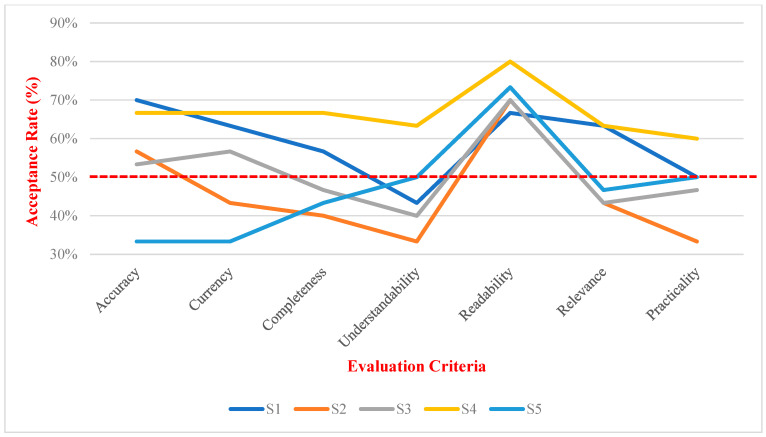
Distribution of ChatGPT’s performance ratings in dietary advice as acceptable by dietitians (>7) for each criterion. Note: The red dashed line indicates the 50% acceptance rate threshold, which serves as a benchmark for evaluating the performance of each indicator. Scenarios: S1: Balanced Eating; S2: Fit Weight; S3: Dining Out Well; S4: Fewer Processed Foods; S5: Limit Sugary Drinks.

**Table 1 nutrients-16-01939-t001:** Achievement level of NL indicators in dietary advice provided by ChatGPT.

	Not Achieved at All	Partially Achieved	Fully Achieved
Scenario (No. of Indicators)	*n* (%)	*n* (%)	*n* (%)
S1: Balanced Eating (8)	137 (57.08)	77 (32.08)	26 (10.38)
S2: Fit Weight (8)	151 (62.92)	71 (29.58)	18 (7.50)
S3: Dining Out Well (5)	65 (43.33)	46 (30.67)	39 (26.00)
S4: Fewer Processed Foods (3)	18 (20.00)	40 (44.44)	32 (35.56)
S5: Limit Sugary Drinks (4)	76 (63.33)	34 (28.33)	10 (8.33)

Note: The evaluation of ChatGPT’s performance was completed by 30 dietitians. The total number of evaluation result items for each scenario is equal to the number of corresponding indicators multiplied by 30.

**Table 2 nutrients-16-01939-t002:** Evaluation of the response quality of ChatGPT’s dietary advice across the five scenarios.

	S1: Balanced Eating			S2: Fit Weight			S3: Dining Out Well			S4: Fewer Processed Foods			S5: Limit Sugary Drinks		
Criteria	Range	M (SD)	Median	Range	M (SD)	Median	Range	M (SD)	Median	Range	M (SD)	Median	Range	M (SD)	Median
Accuracy	4–10	7.90 (1.73)	8	3–10	7.53 (1.94)	8	3–10	7.47 (2.00)	8	2–10	7.90 (1.85)	8	2–10	6.60 (2.14)	7
Currency	3–10	7.70 (1.73)	8	3–10	7.27 (1.91)	7	2–10	7.27 (2.29)	8	2–10	7.73 (1.93)	8	3–10	6.97 (1.69)	7
Completeness	3–10	7.27 (1.89)	8	3–10	6.70 (1.93)	6.5	3–10	6.73 (2.29)	7	3–10	7.90 (1.73)	8	3–10	6.80 (1.96)	7
Understandability	1–10	7.00 (2.20)	7	1–10	6.23 (1.98)	6.5	1–10	6.37 (2.36)	7	3–10	7.40 (1.96)	8	2–10	6.70 (2.25)	7.5
Readability	4–10	8.03 (1.85)	8	4–10	8.03 (1.71)	8	3–10	7.97 (2.04)	8	3–10	8.27 (1.72)	8	3–10	8.00 (1.84)	8
Relevance	1–10	7.43 (2.27)	8	1–10	6.57 (2.49)	7	3–10	6.83 (2.17)	7	3–10	7.83 (1.90)	8	3–10	6.83 (1.95)	7
Practicality	1–10	6.77 (2.37)	7.5	1–10	6.30 (2.14)	7	3–10	6.63 (2.19)	7	3–10	7.67 (1.88)	8	3–10	7.23 (1.91)	7.5

Note: M: mean; SD: standard deviation.

**Table 3 nutrients-16-01939-t003:** The feedback of the dietitians on using ChatGPT-3.5 for dietary advice.

Positive Opinions (Occurrences = 87, 35.95%)	Concerns (Occurrences = 155, 64.05%)
- A02 Provides popularized, preliminary, and easy-to-understand dietary advice (29). - A04 Provides comprehensive and detailed health education information (21). - A01 Provides clear and diverse recommendations in a bulleted format (16). - A03 Provides advice closely related to clinical counseling practice (5). - A08 Provides dietary advice that aligns with the theme (4). - A05 Offers creative strategies for healthy eating (3). - A11 (Compared to web searches) provides more efficient and user-friendly information (3). - A09 Reminds to seek professional advice and consult through official channels (2). - A10 Uses a caring tone to design reminders (2). - A06 Generates information for health education counselors’ reference (1). - A07 Serves as an auxiliary tool for nurturing dietitians (1).	- B02 Provides health education information lacking thoroughness or rigor, leading to misunderstandings or misuse (53). - B07 Health education information is incorrect, especially regarding food categorization (23). - B06 Unable to provide personalized (dietary or exercise) analysis and recommendations (20). - B05 Lacks or inadequately explains portion recommendations (17). - B09 Uses language that does not match local culture or lacks precision (does not adhere to professional terminology) (11). - B11 Content is too superficial to accurately address the needs of the questioner (9). - B9 Information provided has issues of repetition and poor structure (5). - B10 Questioning techniques affect the quality of information generated (5). - B03 Lack of explanation for professional terminology in the nutrition field (4). - B04 Fails to provide explanations or warnings for age groups and groups with special needs (4). - B01 Examples of food items are lacking or insufficient (2). - B12 Excessive content in a single health education session makes it difficult to absorb and implement (2).

Note: A total of 242 feedback entries.

**Table 4 nutrients-16-01939-t004:** Incorrect responses of ChatGPT on an NL test.

Items and Correct Answer	ChatGPT Responses	The Correct Rate of Taiwanese College Students [26]
10. According to Rui’s diet list today ^†^, what do you think about his dietary choices: A. Vegetable servings are just right. B. Too many sugary drinks. C. Calories from fats are too high (T).	B	73.6%
11. “Research shows that being overweight or obese (i.e., BMI ≥ ___) is a major risk factor for chronic diseases such as diabetes, cardiovascular disease, and malignant tumors”. The number in the blank should be ^†^: A. 24 (T) B. 27 C. 30	C	54.2%
14. Rui spends most of his time either attending classes or returning to the dormitory to study. He hasn’t participated in any extracurricular activities and does not have a regular exercise routine. Based on Table ^†^, Rui’s daily calorie requirement should be: A. 2450 kcal B. 2100 kcal C. 1400~1750 kcal (T)	B	54.2%
18. Which of the following products ^†^ has the highest total calorie content? A. Product: Lemon-filled cookies, Weight: 80 g (T) B. Product: Braised beef noodles, Weight: 100 g C. Product: Chocolate milk, Volume: 300 milliliters	B	62.3%
29. Since Rui has never had a habit of exercising in the past, he is just starting to develop an exercise routine. Which of the following exercise plans is more suitable for Rui? A. Take a 40-min walk around the campus after dinner every day with the principle of slight breathlessness and slightly increased heart rate (T). B. Take a 30-min walk around the campus after dinner every day with the principle of no breathlessness and no increased heart rate. C. Engage in more strenuous and breathless exercise at the gym for at least 60 min every day.	B	70.1%
Correct rates of 32 items	84.38%	77.40%

Note: ^†^ 10-provided with the diet list; 11-provided with the BMI classification; 14-provided with the daily activity level, weight, and calorie requirement comparison data; 18-provided with the product calorie labeling for each product.

## Data Availability

The data presented in this study are available upon request from the corresponding author. Due to privacy and ethical concerns, neither the data nor the source of the data can be made publicly available.

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
