# Peer review of "Assessing the Quality of ChatGPT’s Dietary Advice for College Students from Dietitians’ Perspectives"

_nutrients, 2024, doi:10.3390/nu16121939_

Round 1

Reviewer 1 Report

Comments and Suggestions for Authors

Congratulations to this well-conducted study.

My specific comments relate to the following points:

1) Figure 1 needs more elaboration in the main text.

2) How have the prompts be elaborated and how much do they differ in quality and "depth"? It would be interesting to analyse the association between quality of prompt and the responses.

3) Figure 2 needs labelling of the axes. I suppose that the x-axis refers to "n"? But why do the absolute values differ?

4) One minor comment relates to the formatting of the authors and their affiliations which should be adjusted based on the guidelines of the journal.

Author Response

Thank you for your valuable comments. Please refer to the attached file for the responses from the authors.

Reviewer 2 Report

Comments and Suggestions for Authors

Li-Ling and colegues present an interesting work assessing the Quality of ChatGPT's Dietary Advice for College students.

This becomes a modern theme since chat GPT and AI is making quite a sucess all over the world, in multiple areas and this article brings a critical point of view of wrong dietary applications of the chat GPT. The program may be useful for general purposes, but when it comes to personalized aspects, it gives errors.

Even with limitations, I believe that this is an innovative and interesting manuscript.

It wold also be interesting to know the previous opinion of dietitiens of chat GPT and if they were used to work previously with them. In fact, I believe students are much more open minded to AI than adult people and this all may have contributed to a different perception of the results.

Author Response

(The authors gave the same response as above.)

Reviewer 3 Report

Comments and Suggestions for Authors

nutrients-3052113_ Assessing the Quality of ChatGPT's Dietary Advice for College Students from Dietitians' Perspectives

This paper is submitted to the "Nutrition Methodology & Assessment" section of this journal. The aim of the study was to evaluate ChatGPT's performance in providing dietary advice to college students. This evaluation was conducted using a comprehensive approach that included dietitians' perspectives on nutrition literacy (NL) achievement, quality indicators, and an objective NL test. A multidimensional evaluation allows for the assessment of dietary recommendations from both objective and subjective perspectives, identifying potential limitations and areas for improvement. These findings are expected to guide the development of AI-driven dietary counseling tools and emphasize the importance of expert perspectives in evaluating digital health interventions.

The abstract accurately reflects the content of the article. The introduction clearly outlines the problem, supported by relevant literature. The objectives are explicitly stated. The methodology is innovative, presenting a highly interesting multidimensional evaluation.

In the statistical analysis section, it should be indicated whether a sample size calculation was performed for this experiment. Additionally, the descriptive statistics, particularly for the NL test between ChatGPT and college students, should be further developed.

In the results section, specifically in Table 1, I suggest comparing proportions, as this would highlight significant differences, with a predominance of "Not achieved at all," thereby reinforcing the results. A simple comparison of proportions would suffice.

Regarding Table 2, the results would be strengthened if compared. It should be assessed whether they present parametric or non-parametric distribution to determine the most appropriate descriptive methods and tests. This should be detailed in the statistical analysis section.

The discussion comprehensively addresses the topic, evaluating its strengths and weaknesses. It should be noted that further studies on this subject are needed. The conclusions are consistent with the obtained results.

Author Response

(The authors gave the same response as above.)
